# Balloon pressure monitoring for radial artery hemostasis after transradial coronary procedures: protocol for a randomized controlled trial

**Xiaodong Zhang[1‡], Lan Zou[1‡], Dunfu Zhang[1‡], Bangtao Yao[1‡], Junge Chen[1‡], Tianfeng Wei[1], Zhouping Fu[1], Xin Chang[1]\*, Lijuan Chen[1,2]\*, Yan Geng[1]\***

**1** Nanjing Lishui People's Hospital, Zhongda Hospital Lishui Branch, Southeast University, Nanjing, China, **2** Zhongda Hospital, School of Medicine, Southeast University, Nanjing, China

‡ Xiaodong Zhang, Lan Zou, Dunfu Zhang, Bangtao Yao, and Junge Chen contributed equally to this work and should be regarded as co-first authors.
\* changxin@lsrmyy.com (XC); chenlijuan@seu.edu.cn (LC); gengyan2004@163.com (YG)

## Abstract

### Background

Forearm radial artery occlusion (RAO) is a common complication after transradial coronary procedures. Traditional patent hemostasis, relying on operator-dependent assessment, results in labor-intensive processes and inconsistent RAO rates.

### Methods

This is a single-center, prospective, randomized, open-label, parallel-group superiority trial. We plan to enroll 818 patients scheduled for transradial coronary angiography. Participants will be randomly assigned (1:1) to either a novel balloon pressure monitoring system (integrating high-precision digital manometry with physiologically-phased decompression) or traditional patent hemostasis. The primary outcome is the incidence of ultrasound-confirmed forearm RAO at 24 hours post-procedure. Key secondary outcomes include rates of access-site vascular complications and bleeding events, as well as objective metrics of hemostasis efficiency. Recruitment Status: Recruitment commenced in September 2024 and is ongoing; the target sample size is anticipated to be reached by May 2026. Analysis will follow the intention-to-treat principle.

### Results/ Trial Status

As a protocol paper, no results are reported. The trial is currently in the recruitment phase.

**Data availability statement:** No datasets were generated or analysed during the current study. All relevant data from this study will be made available upon study completion.

**Funding:** The funding for the trial received support from Special Research Startup Funding for Introduced Personnel of Nanjing Lishui People's Hospital [grant number: KY07] and Nanjing Health Science and Technology Development Special Fund Project [grant number: ZDXX25202]. The funders had no role in study design, data collection and analysis, decision to publish, or preparation of the manuscript.

**Competing interests:** The authors have declared that no competing interests exist.

## Conclusions

This trial will provide the first large-scale randomized evidence on whether digital manometry-guided compression reduces RAO, potentially bridging the efficacy-effectiveness gap between optimized research protocols and routine practice.

## Trial registration

The trial was registered with the Chinese Clinical Trial Registry (ChiCTR) in August 2024, under the registration number ChiCTR2400088258.

## Introduction

### Background

Transradial access (TRA) is currently the standard vascular access site for coronary angiography and intervention, endorsed by international guidelines due to its superior safety profile over transfemoral access [1]. Advantages include significant reductions in access-site bleeding, vascular complications, and reduction of mortality, especially in higher risk patients [2], along with improved patient comfort and cost-effectiveness [1]. However, forearm radial artery occlusion (RAO) remains a common complication, with real-world incidence up to 33% [3]. This complication compromises the future utility of the artery for subsequent coronary procedures, as a surgical conduit, or for arteriovenous fistula creation [1,2].

Patent hemostasis is the cornerstone for preventing RAO [2]. Yet clinical translation is limited by inability to visualize real-time blood flow, reliance on empirical pressure titration through repeated photoplethysmography [2], and a cumbersome verification process [2]. These drawbacks hinder standardization, consume time and resources, and perpetuate the gap between protocol and practice [2]. Although adjunctive strategies (e.g., ipsilateral ulnar artery compression [4] or distal transradial approach [1,5]) can further reduce RAO, their adoption remains limited by device availability, complexity, and learning curves. An objective, standardized, easily implementable hemostasis system remains an unmet need.

Methodologically, existing evidence is fragmented: most studies are non-randomized or retrospective, with heterogeneity in compression protocols, devices, and RAO definitions [2,3,6–9]. Moreover, the predominance of single-center designs limits generalizability [8–10], further widening the efficacy–effectiveness gap and compromising external validity [11].

To address these challenges, we developed a novel balloon pressure monitoring system that integrates high-precision digital manometry with a physiologically-guided, staged decompression strategy. The present trial is designed to evaluate whether digital manometry-guided compression reduces RAO compared with traditional patent hemostasis, potentially bridging the gap between optimized research protocols and routine practice.

## Methods

### Rationale and theoretical basis

The central hypothesis of this study is grounded in the following pathophysiological principle: effective hemostasis at the radial artery puncture site requires maintaining external compression pressure within a precise "therapeutic window" [2]. The lower limit of this window is the minimum pressure needed to seal the puncture site and prevent bleeding [7], while the upper limit is the maximum pressure that does not completely block antegrade blood flow (thus avoiding blood stasis and endothelial injury) [12].

Traditional patent hemostasis, relying on subjective assessment and imprecise volume-based compression, often fails to maintain pressure within this therapeutic window [2,3]. Consistent with international consensus, RAO prevention relies on three pillars [2]: small sheath/catheter size, adequate intra-procedural anticoagulation (e.g., 6-French sheaths, weight-adjusted heparin to maintain ACT 250–300 s), and non-occlusive ("patent") hemostasis. While the first two are easily standardized, the third remains operator-dependent, representing a key evidence-practice gap [2].

To objectively standardize this critical pillar of radial artery hemostasis, we developed a novel balloon pressure monitoring system. The present trial, a prospective randomized controlled trial conducted in accordance with the SPIRIT guidelines (S1 Checklist, SPIRIT 2025 checklist), compares the safety and efficacy of the novel system versus conventional hemostasis following transradial coronary angiography, with the primary endpoint of preventing RAO. The study timeline and flowchart are illustrated in Figs 1 and 2. The system comprises a radial artery compression balloon, a high-precision digital manometer (± 3.75 mmHg), and a pressure-modulation syringe, as shown in Fig 3. Fig 4 (left panel) outlines the standardized operational protocol for the system, which introduces two key innovations designed to overcome limitations inherent to traditional methods:

**1. Quantitative pressure control: From empirical volume to real-time vascular wall pressure.** The system shifts the control target from the "input volume" of air (mL) to the direct, real-time measurement of the "endpoint metric": the actual pressure exerted on the vascular wall (mmHg). By starting from a supra-systolic safety baseline and titrating down to identify the individual's minimum effective hemostatic pressure, it establishes a precise and reproducible benchmark. This ensures compression is consistently maintained within the individualized therapeutic window, eliminating variability introduced by differences in tissue compliance, adipose thickness, or bandage tightness.

**2. Physiologically phased regulation: Closed-loop management dynamically adapted to hemostasis and repair.** The system's decompression strategy is not governed by rigid time intervals but dynamically synchronizes with the endogenous stages of vascular repair (hemostasis, inflammation, and tissue remodeling) [13], forming an adaptive closed loop (Fig 5). Following initial hemostasis, pressure is cautiously reduced to restore antegrade flow. The resulting shear stress acts as a key biomechanical signal, activating endothelial repair programs—notably by upregulating nitric oxide synthesis [12–15]. This promotes vasodilation and inhibits platelet aggregation, thereby mitigating secondary thrombosis [12–14]. Subsequent stepwise decompression further supports inflammatory clearance and re-endothelialization [14,16]. The entire process adheres to the principle of "increase pressure if bleeding occurs, decrease pressure if hemostasis is secure," ensuring dynamic matching of compression intensity with the patient's real-time hemostatic status.

By integrating these innovations, the balloon pressure monitoring system aims to transform patent hemostasis from an operator-dependent art into a standardized, physiology-guided engineering process, thereby bridging the efficacy-effectiveness gap for RAO prevention.

### Study design and purpose

This is a single-center, prospective, randomized controlled trial (RCT) designed to compare the safety and efficacy of our novel system versus traditional patent hemostasis for post-coronary angiography radial artery hemostasis, with preventing forearm RAO as the primary goal. As the first large-scale RCT focusing on digital manometry-guided

| TIMEPOINT | TRIAL PERIOD | | | | |
|---|---|---|---|---|---|
| | Pre-procedure | Day of Procedure (Post-procedure) | 24 Hours Post-procedure | During Hospital Stay | Optional 30-Day Follow-up |
| **ENROLLMENT:** | | | | | |
| *Eligibility screen* | X | | | | |
| *Informed consent* | X | | | | |
| *Baseline data collection* | X | | | | |
| *Randomization* | | X | | | |
| **INTERVENTION/ COMPARATOR:** | | | | | |
| *Balloon pressure monitoring system* | | X | | | |
| *Traditional patent hemostasis* | | X | | | |
| **ASSESSMENTS:** | | | | | |
| *Procedural data* | | X | | | |
| *Primary Endpoint: Forearm RAO* | | | X | | |
| *Secondary Endpoints: Vascular complications, bleeding events* | | | | X | |
| *Secondary Endpoints: Total hemostasis time, active operator time* | | | | X | |
| *Exploratory Endpoint: 30-day forearm RAO* | | | | | X |
| *Exploratory Endpoint: Operator ease-of-use questionnaire* | | X | | | |
| *Safety assessments: Adverse events, serious adverse events* | | | | X | |
| *Discharge assessment* | | | | X | |

**Fig 1. SPIRIT Schedule of enrollment, intervention, and assessments of the outcomes.** The schedule outlines the timing of participant enrollment, group allocation, post-procedure hemostasis interventions, and the assessment windows for primary and secondary outcomes. RAO, radial artery occlusion.

hemostasis for RAO prevention, this study plans to enroll 818 eligible patients, who will be randomly assigned in a 1:1 ratio to either the balloon pressure monitoring group or the traditional hemostasis group. Allocation concealment will be ensured using computer-generated random sequences and sequentially numbered, opaque sealed envelopes. To

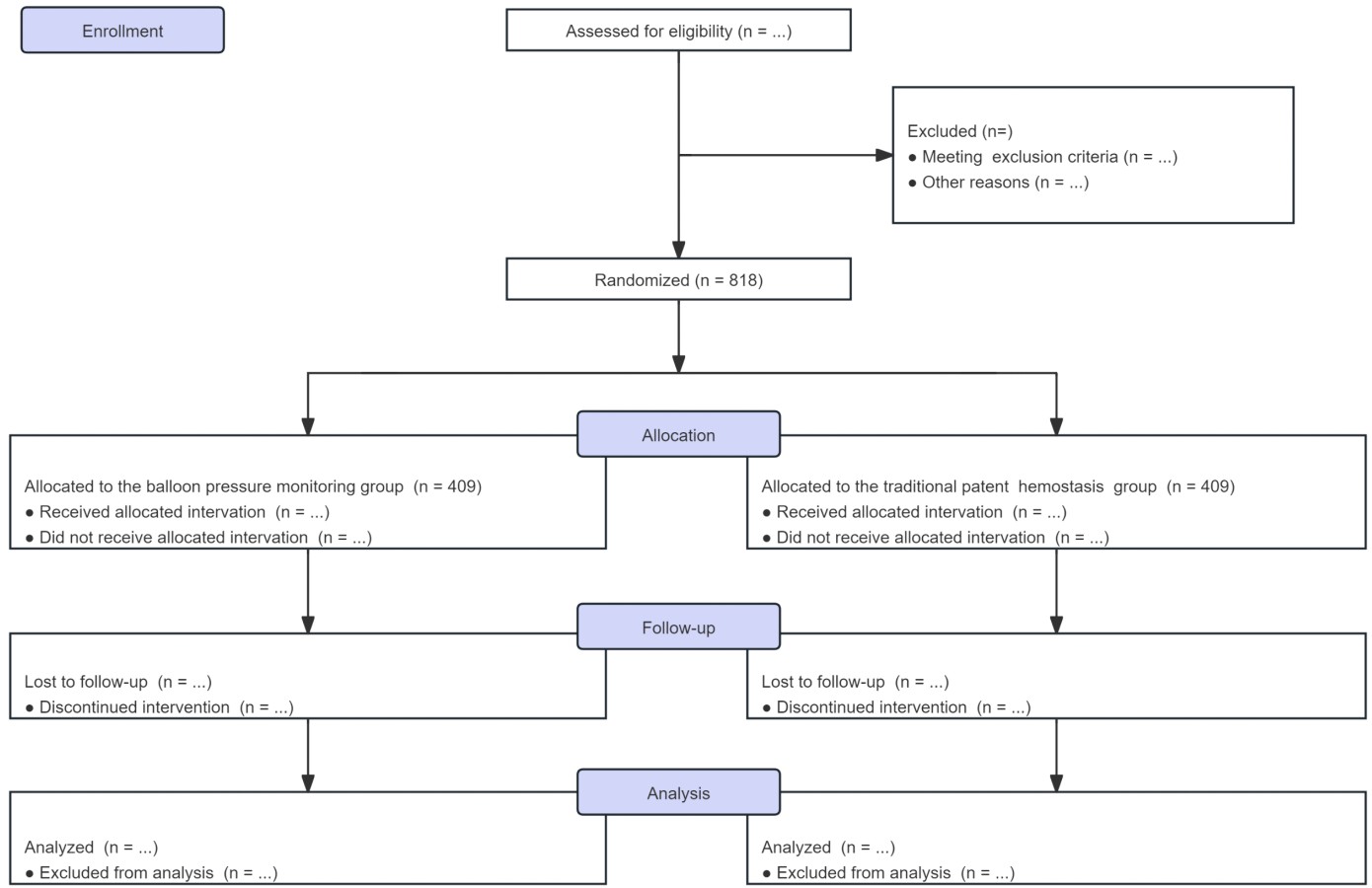

**Fig 2. Study protocol flowchart, adapted from CONSORT.** The flowchart details the screening, randomization, intervention allocation, follow-up, and planned analysis populations.

minimize bias, independent investigators assessing primary and secondary endpoints will be blinded to patient group allocations. Follow-up will continue until hospital discharge, with an optional 30-day outpatient follow-up for exploratory RAO assessment.

## Trial oversight structure

To ensure the trial's scientific rigor, data integrity, and participant safety, the following committees and teams are established, with their composition, roles, and responsibilities detailed below:

   **1. Steering committee.** Composition: The committee comprises the Principal Investigator (PI), co-principal investigators, senior interventional cardiology experts, and a methodology expert.

   Roles & Responsibilities: It is responsible for the overall scientific direction, protocol implementation, and compliance of the trial. It reviews and approves protocol amendments, oversees the operation of subcommittees, and makes decisions on major issues arising during the trial conduct.

   **2. Independent Data and Safety Monitoring Board (DSMB).** Composition: The DSMB consists of two interventional cardiology experts, one independent statistician, and one medical ethicist, all without any conflicts of interest related to this study. The DSMB is completely independent of the trial funders and the study execution team.

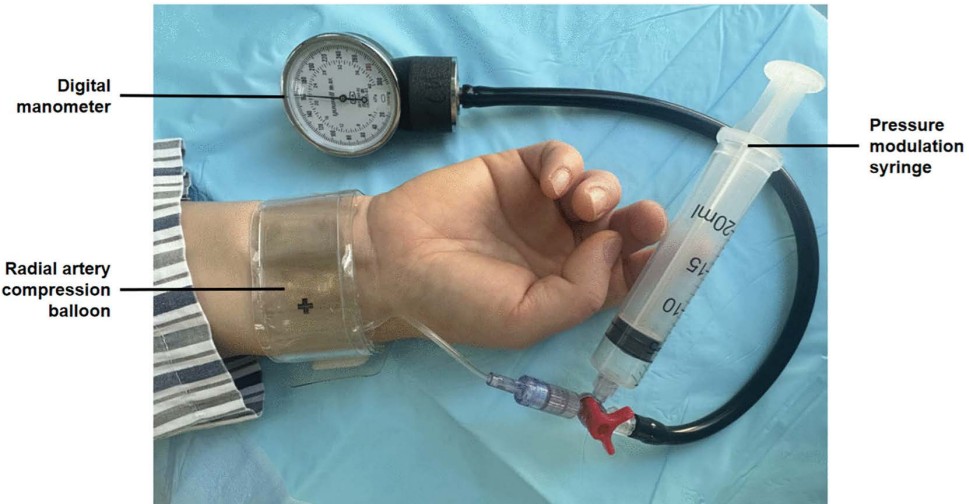

**Fig 3. Structure diagram of the balloon pressure monitoring system components.** The system consists of a radial artery compression balloon, a high-precision digital manometer for real-time pressure feedback, and a pressure-modulation syringe for precise pressure adjustments.

Roles & Responsibilities: In accordance with its independent charter, the DSMB periodically reviews unblinded cumulative safety data (particularly serious adverse events and bleeding events) after every 200 enrolled participants. It assesses the trial's risk-benefit ratio and provides mandatory recommendations to the Steering Committee regarding trial continuation, modification, or termination.

**3. Endpoint adjudication committee.** Composition: This committee includes two vascular ultrasound specialists and one interventional cardiologist, all of whom are blinded to treatment group allocation.

Roles & Responsibilities: It independently adjudicates all primary endpoint events (24-hour radial artery occlusion) and predefined serious vascular complication endpoints (e.g., pseudoaneurysm, arteriovenous fistula). All imaging and clinical data for suspected endpoint events undergo centralized, blinded assessment by this committee based on pre-specified standardized definitions. Its adjudication is final.

**4. Data management team.** Composition: The team consists of the study coordinator, designated research nurses, and a statistician responsible for database management.

Roles & Responsibilities: It is responsible for the design of Case Report Forms (CRFs), data entry, coding, and validation (including double data entry verification and logic checks). The team implements and maintains data security and confidentiality measures. It handles missing data according to the pre-defined statistical plan (see the "Statistical methods" section) and prepares the final dataset for statistical analysis.

**5. Statistical team.** Composition: The team is led by one independent statistician who is blinded to treatment group allocation.

Roles & Responsibilities: The Statistical Team generates the randomization sequence, performs the final statistical analyzes while maintaining the blinding, and provides necessary safety data analyzes to support the DSMB. It reports to the Steering Committee.

## Study timeline and participant recruitment

According to the initial trial registration (ChiCTR2400088258), the planned completion date for participant recruitment was December 31, 2025. However, based on the actual enrollment rate observed since the study commenced in September 2024, it is now anticipated that the target sample size of 818 participants will be reached by May 2026. This adjusted

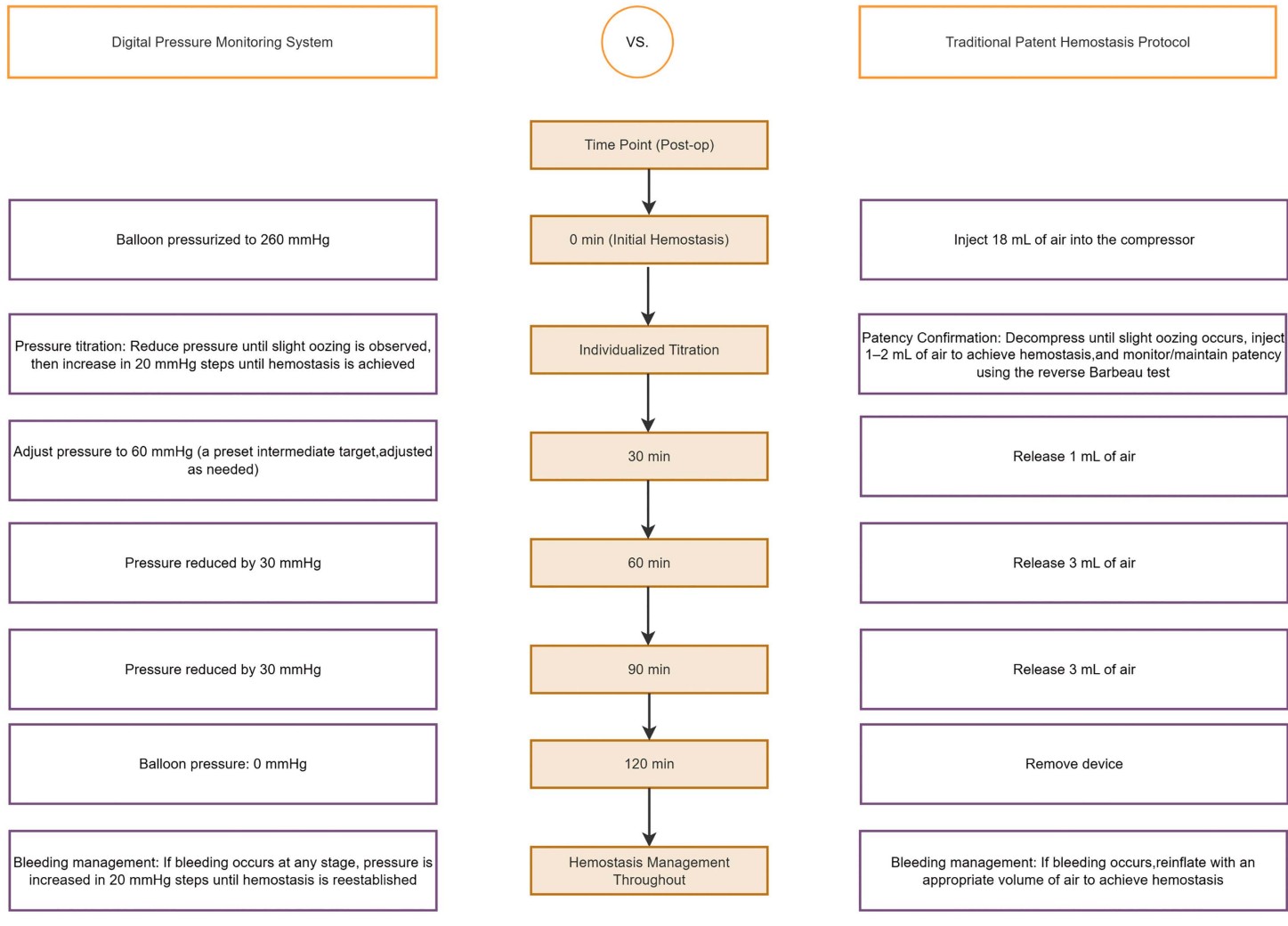

**Fig 4. Comparison of standardized operational procedures.** Left panel: digital pressure monitoring system protocol. Right panel: traditional patent hemostasis protocol.

projection reflects the current pace of recruitment, and the trial registry record will be updated accordingly to reflect this change in timeline.

### Randomization, blinding implementation, and bias control

**Randomization and allocation concealment.** The randomization sequence was generated by an independent statistician using dedicated statistical software. A blocked randomization method was adopted with randomly varying block sizes of 4 and 6, and the block size allocation was concealed from all investigators throughout the trial. Allocation concealment will be maintained via sequentially numbered, opaque sealed envelopes, which will only be opened after patient enrollment and prior to intervention assignment, thus preventing selection bias. The allocation sequence is concealed from investigators involved in enrollment and intervention assignment.

**Blinding implementation.** Given the distinct nature of the two interventions, blinding of the operators performing hemostasis or the patients is not feasible. Therefore, to minimize assessment and measurement bias, the following

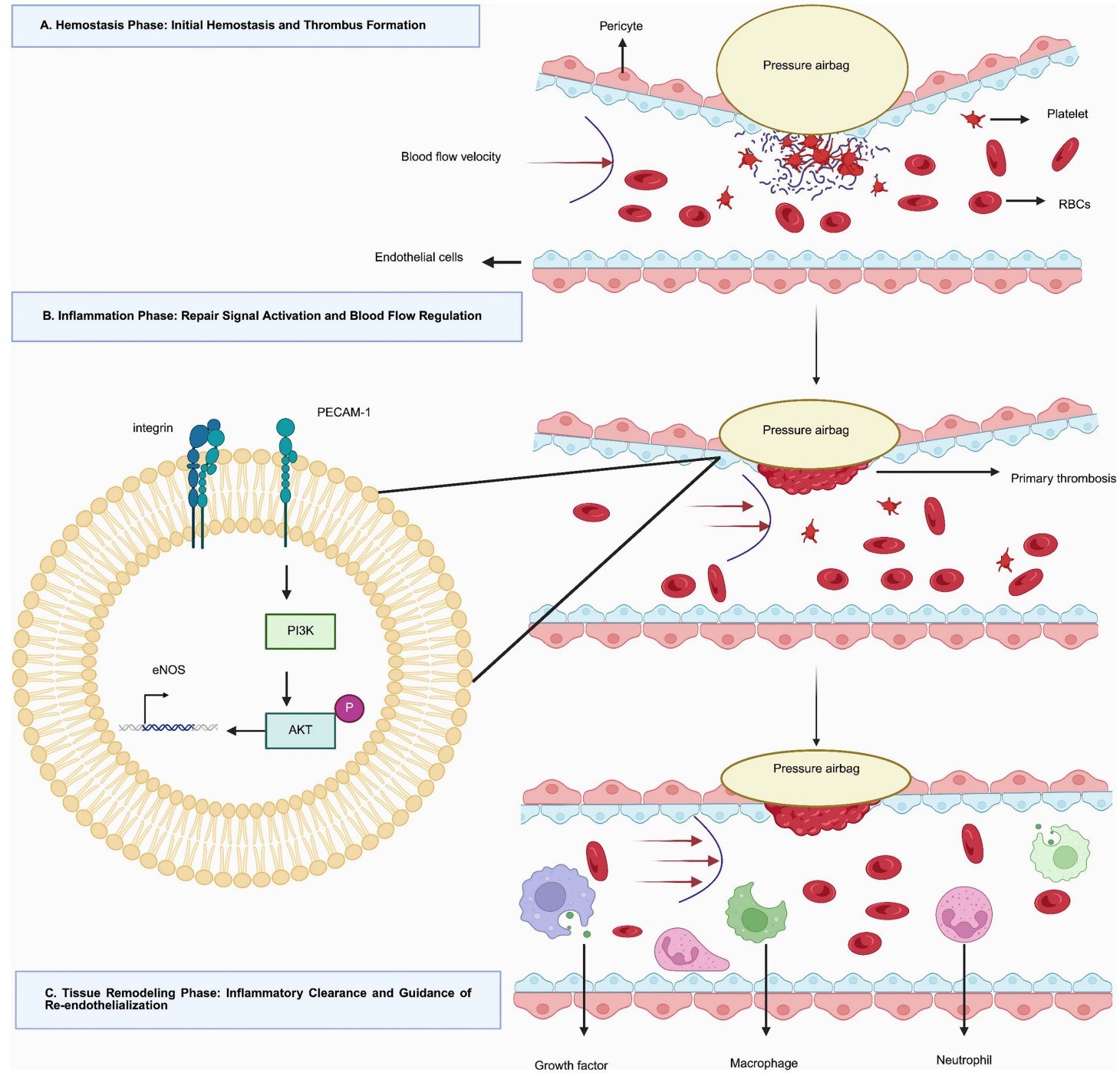

**Fig 5. Mechanism of synchronisation with vascular repair phases.** The diagram illustrates how the system's staged decompression aligns with the three phases of vascular repair: hemostasis/thrombus stabilization (Phase A), endothelial activation/shear-stress-mediated repair (Phase B), and inflammatory clearance/remodeling (Phase C).

blinding measures are implemented: 1) blinding of endpoint assessors, with the primary endpoint (24-hour forearm RAO) assessed by an independent, group-blinded investigator using Doppler ultrasound; and 2) blinding of data analysis, with statistical analysis conducted by an independent statistician blinded to group allocation.

   **Unblinding procedure.** Given that the treating physicians are unblinded, emergency unblinding for clinical management is not required. However, the blinding for the endpoint assessors and statisticians will be maintained throughout the trial. In the exceptional circumstance that unblinding of a participant's allocation is deemed necessary for the safety assessment by the DSMB or for regulatory reasons, a formal request must be submitted to and approved by the Steering Committee. The unblinding will then be performed by the independent statistician who holds the allocation list, and the date and reason will be fully documented.

### Operator training and quality control (integrated for bias mitigation)

Given the unblinded nature of operators, rigorous training and quality control will be implemented to reduce performance and detection bias, as these measures are critical for minimizing bias from non-blinded personnel:

1. Operator qualifications: all staff directly performing balloon pressure monitoring or assessing hemostatic efficacy will be selected from the study team and possess the required clinical credentials and routine work experience, including interventional procedures and post-procedural care.

2. Standardized training: before trial initiation, all relevant personnel must complete a half-day centralized training. This training will cover detailed interpretation of the study protocol and procedures; principles of the system, device calibration, and demonstration of standardized operational workflows; identification and management protocols for endpoint events (e.g., forearm RAO, bleeding, hematoma); and standardized completion of case report forms (CRFs) and data management requirements.

3. Qualification assessment: after training, each operator must successfully complete the full operational workflow for at least three simulated or real patients under the supervision of the PI or a designated supervisor. Formal qualification is granted only after confirming that the operator can execute the protocol accurately and consistently.

4. Ongoing quality monitoring, which will be implemented as follows:

   (a) Periodic review: the PI will regularly audit procedural records and data to ensure protocol adherence.

   (b) Adverse event review: any procedure-related adverse events will trigger team discussions and retraining if necessary.

   (c) Blinded assessment: the primary endpoint (24-hour RAO) will be assessed by a blinded ultrasonographer to eliminate measurement bias.

   (d) Protocol compliance review: an independent team led by the PI will conduct periodic blinded reviews of procedural records, with group allocation concealed, to identify and address deviations.

   (e) Quality control for data collection of key processes: To ensure the consistency and accuracy of the high-frequency recording for key process metrics such as "active operator time," the following measures will be implemented: (i) a designated, trained research nurse will be held solely responsible for timestamp documentation across all cases; (ii) Validated, standardized, and user-friendly timing tools (e.g., dedicated timers or tablet applications) will be used to assist in recording the timestamps of the key operational nodes. The following predefined strategy for handling missing data has already been established in the statistical plan: if the data missing rate is < 10%, a complete-case analysis will be adopted; if the missing rate is ≥ 10%, multiple imputation will be applied in sensitivity analyzes to systematically assess the potential impact of data missingness on the primary study conclusions. (iii) Specialized training for research nurses: Nurses will receive targeted training regarding the definition of "direct hands-on tasks"

                                                                                                                                    

(including pressure adjustment, bleeding assessment, device calibration, and data documentation related to hemostasis; excluding passive observation, waiting, or auxiliary tasks not directly involved in hemostasis). Training will include the case simulations of complex scenarios (e.g., simultaneous multi-operator activities) to ensure consistent identification of task start/end times.

## Common procedural protocol

All patients underwent coronary angiography and/or percutaneous coronary intervention via a transradial approach. A uniform set of pre-procedural and intra-procedural measures, designed to minimize vascular injury and the risk of radial artery occlusion, was applied to all participants before randomization and hemostasis. This standardized protocol was implemented in accordance with contemporary best practices as established in large randomized trials [1].

Arterial Access and Procedure: Vascular access was obtained using a 6-French Glidesheath Slender introducer (Terumo, Tokyo, Japan). The choice of access side (left or right wrist) was at the operator's discretion based on anatomical and clinical considerations. Following sheath insertion, all patients received an intravenous bolus of unfractionated heparin (5,000 IU), with additional boluses administered if needed to maintain an activated clotting time (ACT) between 250 and 300 seconds. To mitigate radial artery spasm, an intra-arterial vasodilator cocktail containing verapamil (5 mg) and/or nitroglycerin (100–200 µg) was administered through the sheath.

Pre-Hemostasis Assessment: Immediately following the coronary procedure and prior to the application of any hemostatic device (and thus prior to group assignment), patency of the access artery was confirmed using the reverse Barbeau test [17]. This step ensured a documented baseline of radial artery flow for all enrolled subjects.

## Criteria for discontinuing or modifying allocated intervention

A participant's allocated intervention (either the novel system or traditional hemostasis) will be discontinued or modified prior to the planned time if: (1) Uncontrollable bleeding or a major vascular complication occurs; (2) The participant experiences intolerable pain or discomfort and requests early device removal; (3) Signs of distal limb ischemia or nerve injury develop; (4) Any other serious adverse event deemed by the treating physician to be related to the hemostasis method and necessitating its cessation.

## Intervention: Operating procedure of the system

The standardized operational procedure for the balloon pressure monitoring system is summarized in Fig 4 (left panel) and detailed as follows:

1. Initial compression: Immediately following sheath removal, the balloon is inflated to a target pressure of 260 mmHg. (This pressure, derived from a pre-trial pilot study (n = 32), ensures reliable immediate hemostasis with no associated neurovascular or skin injuries.)

2. Individualized minimum effective hemostatic pressure titration: The balloon pressure is gradually decreased until slight oozing is observed at the puncture site. Then, the pressure is incrementally increased in steps of 20 mmHg until bleeding ceases. This value is recorded as the patient's initial hemostatic threshold.

3. Phased decompression and monitoring: At 30 minutes post-procedure, the pressure is adjusted to 60 mmHg (an intermediate value based on the hemostatic threshold, with fine-tuning for bleeding status). At 60, 90, and 120 minutes post-procedure, pressure is reduced by 30 mmHg at each interval until it reaches zero.

4. Bleeding management: If bleeding occurs at any stage, pressure is increased in 20 mmHg steps until hemostasis is reestablished.

5. Documentation: All pressure values, operation times, and involved personnel are recorded throughout the procedure to ensure traceability.

(The parameters for this procedure—including the initial pressure, titration step, and decompression schedule—were derived from a preceding pilot study.)

### Control group intervention: Standardized traditional patent hemostasis

**Rationale for comparator selection.** Traditional patent hemostasis was selected as the comparator because it represents the current international consensus and standard-of-care practice for radial artery occlusion prevention following transradial procedures. Comparing the novel balloon pressure monitoring system against this established benchmark allows for a direct assessment of its potential incremental efficacy and safety benefits in a clinically relevant context. To ensure a fair comparison, the control group will undergo a standardized traditional patent hemostasis protocol using a balloon-based radial artery compressor, as detailed below and summarized in Fig 4 (right panel).

**Operational protocol for traditional patent hemostasis.**

1. Initial hemostasis: after sheath removal, inflate the balloon compressor with an initial 18 mL of air for primary hemostasis [2,18].

2. Patency Confirmation (Reverse Barbeau Test): Gradually decompress the balloon until slight oozing is observed at the puncture site, then inject 1–2 mL of air to re-establish the hemostasis. Subsequently, manually compress the ulnar artery for 2 minutes while continuously monitoring the pulse oximetry waveform of the ipsilateral thumb. If the waveform remains present with a stable amplitude throughout the entire compression period (corresponding to Barbeau classification Type A or B), the radial artery patency under compression is confirmed; if the waveform disappears (corresponding to Type D or partial Type C, indicating occlusive compression), immediate further decompression must be conducted until the waveform is restored [2,18].

3. Phased decompression: at 30 minutes post-procedure, release 1 mL of air. At 60- and 90-min post-procedure, release 3 mL of air at each interval. At 120 min post-procedure, remove the compression device [2,18].

4. Bleeding management and documentation: if bleeding occurs, reinflate with an appropriate volume of air to achieve hemostasis, and record the time of intervention and personnel involved [2,18].

### Sample size calculation

This study is designed as a superiority trial to evaluate whether balloon pressure monitoring reduces 24-hour forearm RAO incidence compared with traditional care.

The expected forearm RAO incidence of the control group is set at 4.3%, derived from the control arm (patent hemostasis alone) of the PROPHET-II trial [4]. This rate is consistent with findings from a large meta-analysis (Rashid et al., reporting RAO rates of 3–10% with conventional patent hemostasis) [3] and the international consensus paper (Bernat et al., noting that optimal patent hemostasis in high-volume centers achieves RAO rates of approximately 4–5%) [2]. Thus, the chosen rate represents a rigorous yet clinically realistic benchmark.

The target forearm RAO incidence of the intervention group is ≤ 1.0%, which is the "best-practice" rate achieved via the hemostasis protocols optimized in high-quality trials. This target is informed by the PROPHET-II trial, which demonstrated that a combined strategy of prophylactic ulnar artery compression and patent hemostasis could reduce forearm RAO from 4.3% (at 24-hour follow-up) to 1.0% despite being conducted in high-volume radial centers with standardized protocols. This yields a predefined, clinically meaningful superiority margin of 3.3% (a > 75% relative risk reduction for forearm RAO), which aligns with the efficacy boundaries demonstrated in existing high-level evidence [2].

Regarding trial design selection: We explicitly considered the non-inferiority and equivalence designs, but ultimately opted for superiority design for two core rationales. (1) Mechanistic support: our novel system addresses the fundamental limitations of traditional patent hemostasis (subjective pressure titration, lack of real-time feedback) via high-precision pressure monitoring and physiologically guided decompression, with clear pathophysiological evidence suggesting superior RAO prevention; (2) Clinical value: this study primarily aimed to bridge the efficacy-effectiveness gap by translating the low RAO rates of optimized research into routine practice; a non-inferiority design would only confirm the system is not worse than traditional care, failing to validate its innovative value in reducing RAO incidence. An equivalence design is irrelevant here, as the present study aims only to prove no difference between groups, which contradicts our objective of optimizing outcomes.

Statistical parameters: The sample size was calculated using PASS 15.0.5 software with a two-sided α error of 0.05 and a β error of 0.2 (corresponding to 80% statistical power). Based on a reported 24-hour radial artery occlusion (RAO) incidence of 4.3% in the control group and a predefined superiority margin of 3.3% absolute risk reduction, the initial required sample size was 736 patients with a 1:1 allocation ratio between the two groups. Accounting for an estimated 10% dropout rate, the final planned sample size was set at 818 patients (409 per group).

## Inclusion and exclusion criteria

The inclusion and exclusion criteria (Table 1) were defined to enroll a representative patient population with indications for coronary angiography that reflect routine clinical practice, ensuring the external validity of the trial for real-world application.

## Follow-up plan

Patients will be followed for prespecified endpoints until hospital discharge, with no scheduled post-discharge follow-up for primary endpoints. Consenting patients will be invited to an optional 30-d outpatient follow-up visit for radial artery patency assessment via ultrasound, which will serve as an exploratory endpoint to evaluate the persistence of hemostasis effects and late vascular remodeling.

## Endpoints

**Primary endpoint.** The primary endpoint was set as the incidence of forearm RAO at 24 hours post-procedure—the consensus-recommended timepoint for early patency assessment—assessed by a blinded independent investigator using Doppler ultrasound. Forearm RAO is defined as the absence of an anterograde flow signal in the distal radial artery.

**Table 1. Inclusion and exclusion criteria.**

| *Inclusion Criteria* |
| --- |
| Aged 18 years old or over and less than 80 years old |
| Patients undergoing coronary angiography in the Department of Cardiology |
| Patients with clear consciousness and certain ability to understand and express |
| Patients who are willing to participate in the study and provide written informed consent |
| Use of antiplatelet drugs: 300 mg aspirin enteric-coated tablet + 180 mg ticagrelor tablet or 300 mg Clopidogrel bisulfate tablet one day before surgery; On the day of surgery, 100 mg aspirin enteric-coated tablet + 90 mg ticagrelor tablet or 75 mg Clopidogrel bisulfate tablet. |
| *Exclusion Criteria* |
| Patients with upper limb disability or deformity |
| Patients with local skin or tissue edema or infection |
| Patients with abnormal coagulation function or severe complications who have had two or more ipsilateral radial artery punctures |
| The puncture point is not 1–2 cm below the palm stripe |

**Secondary endpoints.** The secondary endpoints were defined as follows:

1. Incidence of access-site related vascular complications, such as radial artery perforation, arteriovenous fistula, pseudoaneurysm;

2. Incidence of bleeding events, classified per standardized bleeding scales for cardiovascular trials;

3. Objective measures of hemostasis convenience, operationally defined to reduce measurement variability, include: i) total hemostasis time, which is the cumulative duration from arterial sheath removal to complete hemostasis device removal and puncture site hemostasis confirmation, with only active operation and assessment time counted (fixed 30/60/90-min observation periods are excluded to minimize variability); and ii) operator workforce input, which is measured via two metrics (with CRF-designed recording templates for feasibility): peak and active operator number. The peak operator number is the maximum number of healthcare personnel simultaneously performing hands-on tasks (pressure adjustment, bleeding assessment, data recording) at any stage. The active operator time is the sum of durations each operator spends on direct hands-on tasks, recorded via start/stop timestamps for each personnel.

**Exploratory endpoints.** The exploratory endpoints are the incidence of forearm RAO at 30 d post-procedure in consenting patients, and the operator-perceived ease of use, which is assessed via a validated brief questionnaire completed immediately after each hemostasis procedure.

Given the anticipated low incidence of individual major vascular complications, analyzes of these events will be primarily descriptive to document the overall safety profile. All such events will be identified and characterized according to the standardized safety assessment procedures detailed in the Safety Assessment and Harms section.

## Safety assessment and harms

**Definition.** All adverse events (AEs) will be recorded from the time of informed consent until hospital discharge. AEs will be classified as serious (SAEs) according to ICH-GCP guidelines. Bleeding events will be defined and graded according to the Bleeding Academic Research Consortium (BARC) criteria. Vascular complications (e.g., pseudoaneurysm, arteriovenous fistula) will be diagnosed by clinical assessment and confirmed by Doppler ultrasound.

**Assessment.** AEs will be assessed systematically through:

1. Active questioning and physical examination of the access site at predefined time points during hemostasis and prior to discharge.

2. Monitoring of all spontaneously reported symptoms by participants.

3. Review of medical records and laboratory values for any potential signs of harm.

**Causality and adjudication.** The treating physician will initially assess the causality of all AEs in relation to the hemostasis procedure. All SAEs and predefined significant vascular complications will be reviewed and adjudicated by the blinded Endpoint Adjudication Committee (see Trial Oversight Structure).

**Recording.** All AEs will be documented in the electronic case report form (eCRF) using standardized terminology.

## Statistical methods

**Analysis populations.** The populations to be analyzed are as follows:

1. intention-to-treat population, which includes all randomized patients, analyzed according to their original treatment assignment, regardless of protocol deviations or crossovers, for the primary efficacy analysis;

2. per-protocol population, which excludes patients with major protocol violations (e.g., significant hemostasis protocol deviation, use of contraindicated medications) or cross-group interventions, serving as a supportive sensitivity analysis;

3. as-treated population, which comprises patients analyzed according to the actual hemostasis method received, for all safety analyzes.

**Handling of missing data.** If the missing primary endpoint data are less than 5%, all missing values will be clearly reported, with no imputation performed (given the low anticipated rate). In case the missing primary endpoint data are 5–15%, a worst-case scenario sensitivity analysis will be conducted, imputing missing data as events in the intervention group and non-events in the control group, to assess the robustness of the primary conclusion. For over 15% missing data, we will perform multiple imputation using chained equations (MICE) with predictive mean matching (PMM) based on 20 imputed datasets. The imputation model will include baseline covariates (age, sex, diabetes, sheath size, anticoagulation dose) and auxiliary variables associated with missingness. Additionally, the worst-case imputation will also be applied as a sensitivity analysis to assess robustness.

**Descriptive statistics and baseline comparisons.** Continuous variables will be tested for normality using the Shapiro–Wilk test. Normally distributed data will be presented as the mean ± standard deviation and compared via independent samples $t$-test, whereas non-normally distributed data will be presented as the median (interquartile range, IQR) and compared via Mann–Whitney U test. Categorical variables will be presented as frequencies and percentages, with comparisons via the Chi-square test or Fisher's exact test (as appropriate). All tests will be two-sided, with $p < 0.05$ considered statistically significant for baseline comparisons.

**Analysis of the primary and secondary endpoints.** Statistical inference for hypothesis testing will be reserved for the primary endpoint alone, with a two-sided α level of 0.05. All comparisons of the secondary endpoints will be exploratory analyzes; their p-values and effect sizes will be for descriptive purposes and hypothesis generation only, with no formal multiplicity adjustment (e.g., Bonferroni) applied. Exploratory analyzes will also be purely descriptive in nature. Consequently, p-values and effect estimates for both the secondary and exploratory endpoints should be interpreted with extreme caution, and should not be construed as confirmatory evidence of efficacy or safety.

The primary endpoint (24-h forearm RAO incidence) will be compared between groups via Chi-square test (or Fisher's exact test for cell counts < 5). Absolute risk difference, relative risk, and odds ratios (ORs) with 95% confidence intervals (CIs) will be calculated. Comparisons of binary secondary endpoints (e.g., vascular complications, bleeding events) will also follow the same methods. Continuous secondary endpoints (e.g., hemostasis time, active operator time) will be compared using independent samples t-test or Mann–Whitney U test, based on data distribution assessed by the Shapiro–Wilk test.

These intergroup comparisons will constitute the univariate analysis. Variables with $p < 0.10$ in these univariate analyzes, along with clinically relevant baseline characteristics (e.g., age, sex, sheath size, anticoagulation dose), will be considered for inclusion in subsequent multivariate logistic regression models to identify independent factors associated with the outcomes.

**Modeling and adjusted analyzes.** Univariate logistic regression will first identify factors associated with forearm RAO and bleeding complications. The variables set for inclusion in the multivariate logistic regression model are as follows: (1) Variables with $p < 0.10$ in univariate analysis; (2) Clinically relevant baseline characteristics (e.g., age, sex, sheath size, anticoagulation dose) pre-specified based on prior literature and clinical expertise, regardless of their statistical significance in univariate analysis. A stepwise selection method (entry $p < 0.05$, exit $p > 0.10$) will be applied to establish a parsimonious model, with results reported as adjusted ORs (aORs) and 95% CIs. Multicollinearity will be assessed using the variance inflation factor (VIF), with a VIF > 5 indicating significant collinearity and necessitating model refinement.

## Prespecified subgroup and exploratory analyzes

1. **Subgroup analyzes.** For the primary endpoint, subgroup analyzes will be conducted for patient characteristics (e.g., age < 65 vs. ≥ 65 years, sex, diabetes status), procedural factors (e.g., sheath size 5F vs. 6F, diagnostic angiography only vs. percutaneous coronary intervention), and operational factors (e.g., compression time quartiles, minimum hemostatic pressure quartiles).

To assess for any differential treatment effect across subgroups, the interaction between the treatment assignment and each subgroup variable will be tested within a logistic regression model including terms for treatment assignment, the subgroup variable, and their interaction. Continuous variables used to define subgroups (e.g., compression time) will be categorized (e.g., into quartiles) for these analyzes. All subgroup analyzes will be exploratory, and their results should be regarded as only hypothesis-generating. Due to the risk of multiple comparisons and the absence of any formal multiplicity adjustment for interaction tests, definitive interpretations of the findings are not warranted. Continuous variables used to define subgroups (e.g., compression time) will be categorized (e.g., into quartiles) for these analyzes.

2. **Learning curve analysis.** A learning curve analysis will be conducted to assess whether the procedural efficacy and efficiency improve with operator experience using the novel system. The first 50 consecutively enrolled patients in the intervention arm (irrespective of the performing operator) will be defined as the 'early experience' cohort, and will be compared with all subsequent intervention arm patients ('established experience' cohort) on the primary endpoint (forearm RAO incidence) and key procedural efficiency metrics (total hemostasis time). The 50-case cutoff was set based on pre-trial pilot data indicating that operators achieve stable proficiency after approximately 50 procedures [19].

We acknowledge that this cohort-based (rather than operator-based) approach may blend cases from operators at different points in their individual learning curves. Therefore, to evaluate the robustness of the findings, a sensitivity analysis will be conducted by repeating the comparison using only data from the two highest-enrolling operators, with the aim of reducing heterogeneity and provide a more direct assessment of the learning effect within individual operators.

## Interim analysis and stopping guidelines

Given that there is only a single primary endpoint which will be assessed at 24 h, no formal interim efficacy analysis is planned. However, the independent DSMB will conduct periodic reviews of the unblinded safety data (after every 200 enrolled patients) per predefined charter, and may recommend early trial termination only for compelling safety concerns.

## Trial monitoring

Trial conduct will be monitored through a combination of independent safety oversight and internal quality audits to ensure protocol adherence, data integrity, and participant safety.

1. **Independent Safety Monitoring.** The independent DSMB, as described in the Interim Analysis and Stopping Guidelines section, is responsible for safety monitoring. The DSMB will convene after every 200 participants are enrolled to review unblinded safety data.

2. **Internal Quality Audits.** The PI and the study coordination team will conduct internal audits of protocol compliance and data quality on a quarterly basis. Audit procedures will include:

(a) Reviewing a random sample of CRFs against source documents for accuracy and completeness.

(b) Verifying adherence to randomization and intervention assignment procedures.

(c) Assessing the documentation of protocol deviations and informed consent.

(d) Evaluating the completeness of data collection for primary and secondary endpoints.

Findings from both DSMB reviews and internal audits will be formally documented. Any significant issues identified will be reported to the Steering Committee, and corrective actions will be implemented promptly.

## Statistical software

All analyzes will be conducted using validated software (SPSS version 26.0, IBM Corp., Armonk, NY, USA; or R version 4.2.0, R Foundation for Statistical Computing, Vienna, Austria) by an independent statistician blinded to the treatment group allocation.

## Ethical approval and study oversight

This study protocol has been approved by the Medical Ethics Committee of Nanjing Lishui People's Hospital (approval number: 2024KY0726−01) and was registered with the Chinese Clinical Trial Registry (registration number: ChiCTR2400088258). The trial will be conducted in accordance with the principles of the Declaration of Helsinki, and written informed consent will be obtained from all participants prior to enrollment.

An independent DSMB, comprising clinical experts and statisticians not involved in trial conduct, will periodically (after every 200 enrolled patients) review the unblinded safety data to assess the risk-benefit balance and provide recommendations for trial continuation, modification, or termination. The main trial results are expected to be submitted for publication in a peer-reviewed journal by the end of 2026.

## Discussion

### Positioning and innovation of this study in the evidence chain

TRA is the established default strategy for coronary procedures, given its demonstrated advantages in safety and patient outcomes over transfemoral access [1,2]. Despite this, forearm RAO remains the most common procedural complication [1,2], and a significant "efficacy-effectiveness gap" persists: while optimized research protocols [1] can achieve RAO rates below 5% under controlled conditions, the real-world incidence remains as high as 33% [3]. This gap compromises the utility of the radial artery for future interventions, and originates primarily from the limitations inherent in patent hemostasis—the cornerstone of RAO prevention [2,3].

Clinically, patent hemostasis relies on operator-dependent, intermittent assessments and empirical pressure titration, leading to poor standardization, high inter-operator variability, and failure to achieve any true patency in a substantial proportion of real-world patients [2,3]. Methodologically, the existing evidence base is fragmented, characterized by a predominance of single-center or non-randomized designs, significant heterogeneity in devices and protocols, and inconsistent definitions and assessments of RAO [2,6,7]. These limitations obscure the true treatment effects and hinder the translation of research findings into practice [8,9]. Although intensive, multi-component "best-practice bundles" have achieved remarkably low RAO rates in trials, their complexity and resource demands limit their broad implementation, thus underscoring the need for a simplified, objective, and scalable approach to patent hemostasis [2,4].

The innovation of this study lies in addressing the external validity challenge [11]: traditional hemostasis' lack of standardization decouples trial results from clinical practice. Our system bridges this gap via two core innovations to standardize hemostasis: 1) Quantitative pressure control: Replacing empirical volume-based compression with high-precision (± 3.75 mmHg) digital manometry to directly measure vascular wall pressure in real time; 2) Physiologically phased decompression: Synchronizing compression release with vascular repair stages (thrombus stabilization, endothelial activation, inflammatory clearance) via real-time bleeding status (not rigid time intervals).

As the first large-scale randomized controlled trial focusing on a digital manometry-guided hemostasis system for RAO prevention, this study occupies a pivotal position in the evidence chain. It seeks not to replace consensus guidelines, but to enable their reliable implementation by providing the objective control and standardization that traditional methods lack.

We hypothesize that this system could consistently replicate the low RAO rates achieved by complex research protocols by precisely controlling the core biomechanical variable of the vascular wall pressure, thereby bridging the gap between evidence-based recommendation and routine clinical practice with a scalable solution for radial artery preservation.

## Distinctive design features of the present trial

This trial incorporates several key design strengths to ensure robust and clinically relevant evidence:

1. Superiority Design with Clinically Meaningful Endpoint: This trial is powered to detect a pre-specified 3.3% absolute risk reduction in 24-hour RAO (clinically significant, benchmarked against real-world and best-practice targets).

2. Standardization and Bias Mitigation: While operator blinding is not feasible, rigorous measures will be implemented to minimize bias, including standardized training, a unified control protocol, and blinded assessment of the primary endpoint by independent ultrasonographers.

3. Operator Proficiency and Real-World Validity: Operators will be required to complete the structured training and demonstrate competency, thus ensuring the intervention is tested under conditions of proficient use. The broad inclusion/ exclusion criteria ensure a population representative of routine clinical practice.

4. Comprehensive Endpoint Assessment: Beyond the primary efficacy endpoint, the trial systematically evaluates key secondary outcomes, including vascular complications, bleeding events, and (uniquely) objective metrics of procedural burden (e.g., active operator time), providing a holistic view of the system's clinical utility.

## Potential clinical implications and scientific contribution

If this trial demonstrates the superiority of the system, it should provide high-level evidence supporting a paradigm shift toward objective, biomechanically-guided post-procedural care. Clinically, it would offer interventional cardiologists a standardized, user-friendly tool to consistently achieve patent hemostasis, potentially reducing the incidence of forearm RAO closer to the benchmarks set by complex research protocols. This could further solidify the safety profile of the transradial approach and enhance the long-term sustainability for patients requiring repeated vascular access.

From a methodological perspective, this trial advocates for the standardization and objectification of a key procedural step. The success of the study would underscore the importance of precise biomechanical control in vascular injury management and could stimulate innovation in smart, closed-loop hemostasis devices, advancing the field from artisanal skill toward reproducible engineering. Ultimately, by transforming hemostasis from an operator-dependent art into a standardized, physiology-guided procedure, this system should set a new benchmark for reproducible vascular access site management.

## Study limitations

The interpretation of the results of this trial will require consideration of its inherent limitations:

1. Single-Center Design: Conduction of this trial at a high-volume center ensures rigorous protocol adherence and quality control, but may limit the generalizability of the findings to dissimilar centers with lower procedural volumes or different practice patterns. Therefore, the findings of this initial efficacy trial are intended to be applied to a multicenter pragmatic trial (as outlined in the Future Research Directions) to confirm the effectiveness and generalizability across diverse clinical settings.

2. Open-Label Nature: The inability to blind operators may introduce performance bias, primarily via a Hawthorne effect. Awareness of trial participation and procedural monitoring may subconsciously motivate operators to

implement the control-group procedures (traditional patent hemostasis), with a degree of standardization exceeding the routine practice. This could manifest as more common plethysmographic assessments, more meticulous pressure adjustments, or stricter adherence to patency confirmation steps than in usual care. Such behavior may artificially lower the observed rate of RAO in the control group, thus narrowing the measured efficacy difference between groups and potentially underestimating the true superiority of the novel system. Although protocol-mandated standardization—through unified training, procedural checklists, and independent audits—aims to mitigate this risk, the residual performance bias due to the Hawthorne effect cannot be entirely eliminated and will therefore need to be considered when interpreting the results [20].

3. Learning Curve: Although operators are pre-trained, a learning curve associated with the new system is acknowledged and will be quantitatively assessed in a pre-specified analysis.

4. Short-Term Primary Endpoint: The primary endpoint will be assessed at 24 hours. While early forearm RAO is a strong predictor of persistent occlusion, the optional 30-day follow-up has been set as an exploratory endpoint, thereby limiting definitive conclusions regarding the long-term patency. Nevertheless, the 24-hour assessment remains the clinically pivotal and universally adopted endpoint in pivotal RAO prevention trials.

## Future research directions

The findings of this trial should inform the following clearly defined research agenda:

1. Multicenter Pragmatic Trials: To confirm the effectiveness and implementation feasibility across diverse healthcare settings with varying resources and expertise.

2. Health Economic Evaluations: Formal cost-effectiveness analyzes comparing the new system with standard care, thereby incorporating savings from reduced RAO, lower human resource demands, and potential bed-day reductions.

3. Personalized Protocol Optimization: Investigation of the system's performance and potential need for protocol adaptation in key patient subgroups. These will include not only high-risk populations (e.g., elderly, diabetic, or chronic kidney disease patients), but also groups with distinct anatomical or physiological characteristics that could influence the biomechanics of hemostasis, such as obese patients (where tissue composition may affect pressure transmission) and women (who typically have smaller radial artery diameters and may be more sensitive to compression). This research will further explore whether personalized algorithms, adjusted for variables like artery diameter, blood pressure, or body habitus, can further optimize outcomes and facilitate tailored implementation.

4. Long-Term Vascular Outcomes: Extended follow-up studies to assess the durability of radial artery patency and the factors associated with late recanalization or re-occlusion.

## Conclusion

This trial addresses a central paradox in interventional cardiology: specifically, the reliance on a subjective, operator-dependent technique (patent hemostasis) to protect a vessel critical for future access. By evaluating a digitally-monitored, physiology-guided compression system, we aim to demonstrate that objective standardization alone could reliably replicate the low forearm RAO rates of optimized research within real-world practice. Positive results from this trial would redefine the standard of care for radial hemostasis, shifting it from an empirical art toward a controlled engineering process, and provide a foundational evidence base for the next generation of intelligent hemostasis devices.

## Ancillary care, post-trial access, and compensation

This trial does not involve provisions for ancillary or post-trial care. However, in accordance with the Declaration of Helsinki and relevant regulations, the trial sponsor (Nanjing Lishui People's Hospital) will provide necessary medical treatment and cover all reasonable medical costs for any injury determined to be directly caused by participation in this trial.

## Supporting information

**S1 Checklist. SPIRIT 2025 checklist.**
(PDF)

**S2 File. Study protocol V1.1.**
(PDF)

## Acknowledgments

The authors would like to thank the medical staff of the Department of Cardiology and the Department of Diagnostic Ultrasound at Nanjing Lishui People's Hospital for their assistance with the trial.

## Author contributions

**Conceptualization:** Xiaodong Zhang, Lan Zou, Dunfu Zhang, Bangtao Yao, Junge Chen, Xin Chang, Lijuan Chen, Yan Geng.

**Funding acquisition:** Lan Zou, Yan Geng.

**Investigation:** Xiaodong Zhang, Lan Zou, Dunfu Zhang, Bangtao Yao, Junge Chen, Tianfeng Wei, Zhouping Fu.

**Methodology:** Xiaodong Zhang, Lan Zou, Dunfu Zhang, Bangtao Yao, Junge Chen, Xin Chang, Lijuan Chen, Yan Geng.

**Project administration:** Xin Chang, Lijuan Chen, Yan Geng.

**Supervision:** Xin Chang, Lijuan Chen, Yan Geng.

**Writing – original draft:** Xiaodong Zhang, Lan Zou, Dunfu Zhang, Bangtao Yao, Junge Chen.

**Writing – review & editing:** Tianfeng Wei, Zhouping Fu, Xin Chang, Lijuan Chen, Yan Geng.

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
