## [Decision Letter · Decision Letter 0]

14 Apr 2026

PONE-D-26-00262Balloon Pressure Monitoring for Radial Artery Hemostasis After Transradial Coronary Procedures: Protocol for a Randomized Controlled TrialPLOS One

Dear Dr. Geng,

Thank you for submitting your manuscript to PLOS ONE. After careful consideration, we feel that it has merit but does not fully meet PLOS ONE’s publication criteria as it currently stands. Therefore, we invite you to submit a revised version of the manuscript that addresses the points raised during the review process.

Please address each reviewer comment individually in your response letter, and we thank you for your submission.

We look forward to receiving your revised manuscript.

Kind regards,

R. Jay Widmer

Academic Editor

PLOS One

Journal Requirements:

Additional Editor Comments:

The reviewers were generally favorable on this protocol manuscript, with just a few items to consider revising in both the paper and the overall project plan. We would encourage you to shorted the introduction and really optimize the description of the methods and statistics. Please address each comment individually in your response letter. Thank you.

Reviewers' comments:

Reviewer's Responses to Questions

**Comments to the Author**

1. Does the manuscript provide a valid rationale for the proposed study, with clearly identified and justified research questions?

Reviewer #1: Yes

Reviewer #2: Yes

2. Is the protocol technically sound and planned in a manner that will lead to a meaningful outcome and allow testing the stated hypotheses?

Reviewer #1: Yes

Reviewer #2: Yes

3. Is the methodology feasible and described in sufficient detail to allow the work to be replicable?

Reviewer #1: Yes

Reviewer #2: Yes

4. Have the authors described where all data underlying the findings will be made available when the study is complete?

Reviewer #1: Yes

Reviewer #2: Yes

5. Is the manuscript presented in an intelligible fashion and written in standard English?

Reviewer #1: Yes

Reviewer #2: Yes

6. Review Comments to the Author

You may also provide optional suggestions and comments to authors that they might find helpful in planning their study.

Reviewer #1: Interesting paper. Some issues should be added

1) introduction should be shortened to no more than a page

2) is there a patent for this sytem?

3) open lable design is a limitation and should be added

4) kind of multiple imputation should be added (it should be overrealistic)

5) single center design limits generazibility

6) for sample size calculation data derive incidence of events from a study describing high volume center in optimal conditions. Please quote other papers

7) number of patients in each block shiuld be added

Reviewer #2: This manuscript presents a well-structured and methodologically sound trial protocol. The research question is relevant, and the study design is appropriate and clearly described. The level of detail provided ensures transparency and reproducibility.

Strengths:

The manuscript demonstrates several strengths, including a robust methodological framework, a clearly defined statistical analysis plan, and appropriate ethical oversight. The intervention and outcomes are well justified, and the protocol follows established reporting standards.

Comments:

I have no major concerns regarding the study design or reporting.

7. PLOS authors have the option to publish the peer review history of their article (what does this mean?). If published, this will include your full peer review and any attached files.

Reviewer #1: **Yes:** Fabrizio D'Ascenzo

Reviewer #2: **Yes:** Pedro Beraldo de Andrade

---

## [Author Response · Author response to Decision Letter 1]

11 May 2026

Response to Reviewer 1

We thank the reviewer for the insightful comments. We have addressed each point as follows and have revised the manuscript accordingly. All changes are highlighted in the revised manuscript using track changes.

Comment 1

introduction should be shortened to no more than a page

Response

We agree with the reviewer. The introduction has been substantially shortened to no more than one page. We have removed redundant content and streamlined the narrative while preserving all key background, evidence gaps, and study rationale.

Comment 2

is there a patent for this system?

Response

We appreciate your question regarding the patent status of the balloon pressure monitoring system. We have added a clear statement in the Competing interests section that the balloon pressure monitoring system described in this protocol is not protected by any patent. No patent is associated with this system.

Comment 3

open label design is a limitation and should be added

Response

We appreciate the reviewer’s careful comment. We would like to clarify that the open-label design has already been clearly recognized and fully discussed as an important limitation in the Study Limitations section of the manuscript. We have detailed the potential performance bias, Hawthorne effect, and related impacts introduced by the open-label design, together with the bias-mitigating strategies adopted. The relevant description has been maintained unchanged in the revised manuscript.

Comment 4

kind of multiple imputation should be added (it should be overrealistic)

Response

We thank the reviewer for the constructive suggestion. We have revised the handling of missing data to specify the type of multiple imputation (MICE with PMM) and added relevant details of the imputation strategy as recommended.

Comment 5

single center design limits generazibility

Response

We appreciate the reviewer’s comment. The limitation of the single-center design and its impact on generalizability has already been clearly stated and fully discussed in the Study Limitations section. The relevant content has been retained unchanged in the revised manuscript.

Comment 6

for sample size calculation data derive incidence of events from a study describing high volume center in optimal conditions. Please quote other papers

Response

We thank the reviewer for the valuable suggestion. We have supplemented the sample size justification with additional citations from a large meta analysis and an international consensus paper to support the control group event rate, while keeping all other parameters and calculations unchanged.

Comment 7

number of patients in each block should be added

Response

We apologize for the omission. We have now added the block size information in the randomization section. The sequence uses randomly varying block sizes of 4 and 6, which are concealed from investigators.

Response to Reviewer 2

We greatly appreciate your positive feedback and constructive comments on our manuscript. We are pleased that you found the trial protocol well-structured, methodologically sound, transparent, and reproducible. All comments from the other reviewer have been carefully addressed and revised in the manuscript accordingly. Thank you again for your time and valuable input.

---

## [Decision Letter · Decision Letter 1]

15 May 2026

Balloon Pressure Monitoring for Radial Artery Hemostasis After Transradial Coronary Procedures: Protocol for a Randomized Controlled Trial

PONE-D-26-00262R1

Dear Dr. Geng,

We’re pleased to inform you that your manuscript has been judged scientifically suitable for publication and will be formally accepted for publication once it meets all outstanding technical requirements.

Kind regards,

R. Jay Widmer

Academic Editor

PLOS One

Additional Editor Comments (optional):

We thank the authors for completing all requested revisions.

Reviewers' comments:

Reviewer's Responses to Questions

**Comments to the Author**

1. Does the manuscript provide a valid rationale for the proposed study, with clearly identified and justified research questions?

Reviewer #1: Yes

2. Is the protocol technically sound and planned in a manner that will lead to a meaningful outcome and allow testing the stated hypotheses?

Reviewer #1: Yes

3. Is the methodology feasible and described in sufficient detail to allow the work to be replicable?

Reviewer #1: Yes

4. Have the authors described where all data underlying the findings will be made available when the study is complete?

Reviewer #1: Yes

5. Is the manuscript presented in an intelligible fashion and written in standard English?

Reviewer #1: Yes

6. Review Comments to the Author

You may also provide optional suggestions and comments to authors that they might find helpful in planning their study.

Reviewer #1: all comments have been satisfied and authors should be complimented for performing such a rigorous revision.

7. PLOS authors have the option to publish the peer review history of their article (what does this mean?). If published, this will include your full peer review and any attached files.

Reviewer #1: **Yes:** Fabrizio D'Ascenzo

---

## [Editor Report · Acceptance letter]

PONE-D-26-00262R1

PLOS One

Dear Dr. Geng,

I'm pleased to inform you that your manuscript has been deemed suitable for publication in PLOS One. Congratulations! Your manuscript is now being handed over to our production team.

Kind regards,

on behalf of

Dr. R. Jay Widmer

Academic Editor

PLOS One